# Mark Burgin's Legacy: The General Theory of Information, the Digital Genome, and the Future of Machine Intelligence

## Rao Mikkilineni

Opos.ai, Chief Technology Officer, San Mateo, CA 94401, USA; rao@opos.ai

**Abstract:** With 500+ papers and 20+ books spanning many scientific disciplines, Mark Burgin has left an indelible mark and legacy for future explorers of human thought and information technology professionals. In this paper, I discuss his contribution to the evolution of machine intelligence using his general theory of information (GTI) based on my discussions with him and various papers I co-authored during the past eight years. His construction of a new class of digital automata to overcome the barrier posed by the Church–Turing Thesis, and his contribution to super-symbolic computing with knowledge structures, cognizing oracles, and structural machines are leading to practical applications changing the future landscape of information systems. GTI provides a model for the operational knowledge of biological systems to build, operate, and manage life processes using 30+ trillion cells capable of replication and metabolism. The schema and associated operations derived from GTI are also used to model a digital genome specifying the operational knowledge of algorithms executing the software life processes with specific purposes using replication and metabolism. The result is a digital software system with a super-symbolic computing structure exhibiting autopoietic and cognitive behaviors that biological systems also exhibit. We discuss here one of these applications.

**Keywords:** digital genome; general theory of information; structural machines; machine intelligence; knowledge structures; cognizing oracles; super-symbolic computing





## 1. Introduction

The contribution of Mark Burgin's general theory of information (GTI) provides a bridge between our understanding of the material world consisting of matter and energy and the mental worlds of biological systems which utilize information and knowledge [1–3] to interact with the material world. Biological systems, while made up of material structures, are unique in their ability to maintain the identity of their structures, observe themselves and their interactions with the external world using information processing structures, and make sense of what they are observing fast enough to do something about it while they are still observing it. Without this ability, their structural stability, sustenance, safety, and survival are at stake. All living beings, whether they are in the form of metabolic networks, interactive networks, or the networks of genes, and neurons [4,5], have developed the ability to sense and act using metabolism (the ability to use the transformation of matter and energy) to minimize the entropy of their structures by exchanging energy with their environment. In addition, through natural selection, they have developed the ability to transmit their knowledge through replication from the survivor to the successor. The biological structures are self-regulating and use autopoietic and cognitive processes, which assure their stability, sustenance, safety, and survival through homeostasis. The genome provides the operational knowledge to execute life processes used to build, self-organize, operate, and maintain the system using both inherited and learned knowledge and assure stability, sustenance, safety, security, and survival in the face of fluctuations in the interactions within the system and with its environment.

As described by Itai and Lercher in their book [6] (p. 11) *The Society of Genes*, the single fertilized egg cell develops into a full human being is achieved without a construction manager or architect. The responsibility for the necessary close coordination is shared among the cells as they come into being. It is as though each brick, wire, and pipe in a building knows the entire structure and consults with the neighboring bricks to decide where to place itself".

The single cell replicates into trillions of cells, each executing a process with a purpose using metabolism and sharing information with other cells to execute a hierarchy of processes to manage and maintain life as defined in the genome. These processes execute autopoietic and cognitive behaviors. The autopoietic behaviors are capable of regenerating, reproducing, and maintaining the system by itself with the production, transformation, and destruction of its components and the networks of processes in these components. The cognitive behaviors are capable of sensing, predicting, and regulating the stability of the system in the face of both deterministic and non-deterministic fluctuations in the interactions among the internal components or their interactions with the environment.

This is in contrast to an assembly of material structures that are composed into a complex adaptive system that undergoes emergence and phase transition under the influence of large fluctuations in the interactions of the components and the environment without self-regulation and the ability to control their destiny [7,8]. The outcome of emergence is unpredictable and the system has no influence on the final state which depends entirely on the system components, their functions, and the composed structure which is influenced by the fluctuations in their interactions. Entropy and energy determine the fate of the system's state under large fluctuations. Biological systems, on the other hand, have self-awareness and the ability to maintain their structural stability through homeostasis. As mentioned earlier, they are able to observe themselves and their interactions with their environment and make sense of what they are observing fast enough to do something about it while they are still observing it. The genome imparts this ability to process information through the operational knowledge to execute life processes transmitted by the survivor to the successor. The successors use the knowledge that is both inherited and learned through life processes to maintain homeostasis and control their destiny when faced with fluctuations in both internal and external interactions. The mental world of the biological system is created by the cognitive structures it develops to process the information it receives through its senses interacting with the material structures. Thus, information is the bridge between the material and mental structures.

GTI provides a framework for understanding the nature of the observer and the observed using information as the bridge between the material and mental worlds. In this paper, we discuss the relationship between the material and mental worlds using the GTI, its use in understanding how intelligence in biological structures is related to their information processing structures, and how to use this knowledge and infuse intelligent behaviors into digital automata. We will conclude this paper with the observation that Mark Burgin's contribution will have a profound impact on how we build, operate, and maintain next-generation information systems with the ability to observe themselves and their interactions with the external world using digital information processing structures and make sense of what they are observing fast enough to assist humans and other machines in doing something about it while they are still observing it. This ability goes beyond the current state-of-the-art information systems using symbolic and sub-symbolic computing alone as practiced today. The super-symbolic computing structures suggested by Burgin [9] using the structural machine and knowledge structures derived from GTI provide an overlay over symbolic and sub-symbolic computing structures with a common knowledge representation. This is very similar to the development of the neocortex integrating the knowledge received from the cortical columns to develop a higher level of reasoning.

In this paper, we will summarize the GTI and its impact on future information processing structures. In Section 2, we provide an introduction to GTI and the role of information as a bridge between the observer (mental world) and the observed (the material world). In

Section 3, we discuss the role of epistemic information and knowledge processed by the mental structures of biological systems using their cognitive apparatuses. In Section 4, we briefly discuss the schema-based digital automata and the use of a digital genome to build a new class of autopoietic and cognitive machine intelligence. In Section 5, we describe an implementation of a medical knowledge-based digital assistant using the design of a digital genome addressing the process of early diagnosis. In Section 6, we discuss the ramifications of the new approach on the current state of information systems design using symbolic and sub-symbolic computing structures. We conclude with Section 7, presenting the final thoughts on the impact of the Late Prof. Mark Burgin's writings and some future research directions for the next generation of computer scientists and information technology professionals.

## 2. The General Theory of Information Providing a Bridge between the Material and Mental Worlds

One of the contributions of Burgin is to provide a scientific interpretation of Plato's Theory of Ideas [10] (p. 1) using the theory of ideal structures consisting of fundamental triads/named sets. "For millennia, the enigma of the world of Ideas or Forms, which Plato suggested and advocated, has been challenging the most prominent thinkers of humankind. This paper presents a solution to this problem, namely, that an Idea in Plato's sense can be interpreted as a scientific object called a structure. To validate this statement, this paper provides a rigorous definition of a structure and demonstrates that structures have the basic properties of Plato's Ideas. In addition, we describe the world of structures and prove its existence".

GTI is a unique theory that relates the material structures in the physical world and the mental structures that biological systems use to model their observations and interact with their external environment using the cognitive apparatuses they build from the knowledge inherited from the genome passed on by the survivors to their successors. The scientific object called the structure is used to provide a schema and operations to represent the knowledge that the biological systems create or update using the information received through their five senses. According to GTI [1,2,11] and the theory of structural reality [3], "knowledge is related to information as the matter is related to energy." Energy in the physical world has the potential to create or change the structure of matter. Material structures interact with each other obeying the laws of transformation of energy and matter. They carry ontological information that represents their state and the dynamics (or the evolution of the system in time) of the structure under consideration in the material world. Figure 1 summarizes the relationship between material structures, mental structures, and ideal structures.

The physical universe, as we know it, is made up of structures that deal with matter and energy. Energy has the potential to create or modify physical structures. Material structures interact with each other and their state and dynamics are subject to laws of nature. GTI tells us that the information about the state and the dynamics carried by the material structures is received by the biological systems through their senses and is converted into knowledge which helps them to create mental structures. Information has the potential to create or modify knowledge and update mental structures. Information is the bridge between the material world and the mental world. GTI also tells us that both the material and mental structures are represented using the ideal structures in the form of fundamental triads/named sets.

A genome in the language of GTI [1,12] encapsulates "knowledge structures" [13] coded in the form of DNA and is executed using the "structural machines" [14–17] in the form of genes and neurons which use physical and chemical processes (dealing with the conversion of matter and energy). The information accumulated through biological evolution is encoded into knowledge to create the genome which contains the knowledge network defining the function, structure, and autopoietic and cognitive processes to build and evolve the system while managing both deterministic and non-deterministic fluctu-

ations in the interactions among the internal components or their interactions with the environment. In the next section, we will discuss how information and knowledge are related to intelligence in biological systems and discuss its relevance to machine intelligence as pointed out by Burgin [18].

**Figure 1.** Material structures and mental structures are represented by ideal structures that describe the state and its dynamics [4].

### 3. Data, Information, Knowledge, and Intelligence

As Burgin [19] points out, GTI explains the relevant relations between information, knowledge, and data demonstrating that while knowledge and data are objects of the same type with knowledge being more advanced than data, information has a different type.

Burgin developed a mathematical theory [20] (p. 47) of knowledge, "explicating a hierarchy of data types that exists between raw data and knowledge. The aim is to construct such a theory that expresses basic invariant properties of knowledge, which are independent of knowledge representation. In addition, this theory allows us to describe relations between knowledge, data, and information, demonstrating that the conventional approach lacks many important aspects, is too inexact, and thus, misleading in some cases. This research is oriented on the utilization of the obtained results for developing huge distributed knowledge bases and efficient computer knowledge processing. The mathematical apparatus for the theory of knowledge is taken primarily from the theory of named sets and the theory of abstract properties".

In essence, data are mental or physical observable entities with a name and value given by the observer using the learning processes inherited from the genome. Information depicts the relationships between various observed entities. Knowledge includes the dynamics of the state of the system containing a set of entities, their relationships, and the dynamics that define the state's history based on various interactions that change the state of the system. The contribution of GTI is to provide a knowledge representation using ideal structures in the form of fundamental triads/named sets depicting the entities, relationships, and behaviors when events change the state. Figure 1 contains the knowledge representation (a knowledge structure in the language of GTI) derived from the epistemological information. A knowledge structure contains the operational knowledge that changes the state of the system. This operational knowledge defines a process that describes the state of the system and its evolution. Intelligence is the ability to use knowledge to execute a process with a specific goal and biological systems have developed various

mental processes to execute these goals. Intelligence belongs to self-regulating systems with an awareness of the "self", its interactions with its environment, and a purpose that defines the end state and actions that achieve it to change the current state.

According to Burgin [18], it is possible to distinguish three ways of defining intelligent systems. The first approach implies that a system is intelligent when it can solve some complex problem or carry out some complex activity in a simple environment. It is called local intelligence. A process that defines the current state, a goal, and an algorithm that reaches the goal fits this description. A plant cell with a well-defined process converts sunlight into chlorophyll. The second approach considers an intelligent system as a system that can function well in a definite domain, which demands solving a group of (complex) problems. It is called (advanced) cluster intelligence. A group of cells working together with shared knowledge and communicating with information exchange using electrical or chemical signals is an example of group intelligence. Burgin points out that "it is necessary to understand that cluster intelligence is not always higher than local intelligence. For instance, by contemporary measures, cluster intelligence that can solve a group of problems by performing arithmetical operations is lower than local intelligence that can play chess on the level of a world champion". The third approach assumes that a system is intelligent when it can efficiently function in complex conditions or successfully survive in a hostile environment. It is called global intelligence and is used to fulfill system-wide goals using multiple groups providing cluster intelligence. Global intelligence requires system-wide self-regulation to manage cluster intelligence by using addressing, alerting, supervision, and mediation services to optimize the system-wide goals while balancing the constraints at local and cluster levels. These include managing contention for resources, prioritizing tasks based on global requirements, etc.

Intelligence is the ability to acquire and apply knowledge and skills. It is the ability to learn from experience, adapt to new situations, understand and handle abstract concepts, and use knowledge to manipulate one's environment. Intelligence is not a cognitive or mental process per se but rather a selective combination of these processes with a purpose directed toward effective adaptation. It is important to recognize that intelligence requires a goal, the operational knowledge to accomplish that goal, and a means to acquire the resources and act. In the next section, we discuss how GTI provides a path to model not only the autopoietic and cognitive behaviors in the biological systems that contribute to intelligent behavior [12] but also how to infuse these behaviors into digital automata in the digital world [21].

## 4. Burgin–Mikkilineni Thesis and the Digital World

Human intelligence has allowed us to create a digital world based on silicon. The software and hardware which started from a simple observation by Alan Turing [22] that a "man in the process of computing numerical functions can be replaced by a machine that is capable of only a finite number of simple operations", has become a powerful means for advanced machines, automating business processes, and mimicking behaviors learned through processing big data. The Turing machine schema allows data to be represented as sequences of symbols and information relating to the state of a system (entities and relationships) to be represented by data structures. The evolution of the dynamics of the system is described as algorithms (or a series of tasks) also represented by another sequence of symbols that operate on the data structures. The operations are carried out using a CPU, memory, and power (energy) required to operate them.

The Church–Turing thesis is a hypothesis about the nature of computable functions, which states that "a function on the natural numbers is computable by a human being following an algorithm, ignoring resource limitations, if and only if it is computable by a Turing machine". As Cockshott et al., point out, "The concept of the universal Turing machine has allowed engineers and mathematicians to create general-purpose computers and "use them to deterministically model any physical system, of which they are not themselves a part to an arbitrary degree of accuracy. Their logical limits arise when we

try to get them to model a part of the world that includes themselves" [23] (p. 215). This brings up the limitations of the Turing machine to infuse autopoietic and cognitive behaviors that mimic the behaviors of biological systems which require an integrated model of the computer and the computed. This was pointed out by Burgin [13] and he proposed knowledge structures, cognizing oracles, and structural machines as a way to infuse autopoietic and cognitive behaviors into digital automata [13–17].

The Burgin–Mikkilineni (BM) thesis is a hypothesis that deals with the autopoietic and cognitive behavior of artificial systems. According to the ontological BM thesis, the autopoietic and cognitive behavior of artificial systems must function on three levels of information processing systems and be based on triadic automata [4,12,14,24]. The BM thesis is relevant to the limitations of the Church–Turing thesis in that it addresses the challenges posed by rapid non-deterministic fluctuations that drive the demand for resource readjustment in real time without interrupting service transactions in progress. The autopoietic and cognitive behaviors of artificial systems, as described by the BM thesis, provide a solution to managing these challenges without disrupting the computation itself.

As discussed earlier, any algorithm that can be specified is made into an executable function using CPU and memory. Functions operate on data structures and the computation evolves from the current state to a new state without any regard to the state's past history (Markovian process). As long as there are enough resources (CPU and memory), the computation will continue as encoded in the algorithm. It is interesting to note that the Turing computable functions also include algorithms that define neural networks, which are used to model processes that cannot be described themselves as algorithms such as voice recognition, video processing, etc. Cognition here comes from the ability to encode how to mimic neural networks in the brain model and process information just as neurons in biology do.

In summary, GTI provides a framework with structural machines, cognizing oracles, and knowledge structures to model autopoietic and cognitive behaviors. The schema and the operations on the schema provide a means to represent process knowledge and execute the processes in a multi-layered network using the fundamental triads or named sets in the form of entities, relationships, and event-induced behaviors of the system. A knowledge structure defines various triadic relationships between all the entities that are contained in a system. The knowledge structure provides the schema and various operations to evolve the schema from one state to another. Various instances of the knowledge structure schema are used to model the domain knowledge and process information changes as they evolve with changes in their entities and their attributes and behaviors. Knowledge structures, therefore, integrate the dynamics of the system with the static data structures representing its state. The structural machine is an information processing structure that represents the knowledge structures as schema and performs operations on them to evolve information changes in the system from one instant to another when any of the attributes of any of the objects change.

Figure 2 shows the relationship between the ideal structures and the physical, mental, and digital worlds. Biological systems use genes and neurons to convert information into knowledge. The neurons fired together wire together to represent knowledge. The neural networks wired together fire together to execute autopoietic and cognitive behaviors.

In the next section, we will discuss the digital genome and its use in implementing a medical knowledge-based digital assistant that executes an early diagnosis process using knowledge structures, cognizing oracles, and structural machines [4,13–16,25].

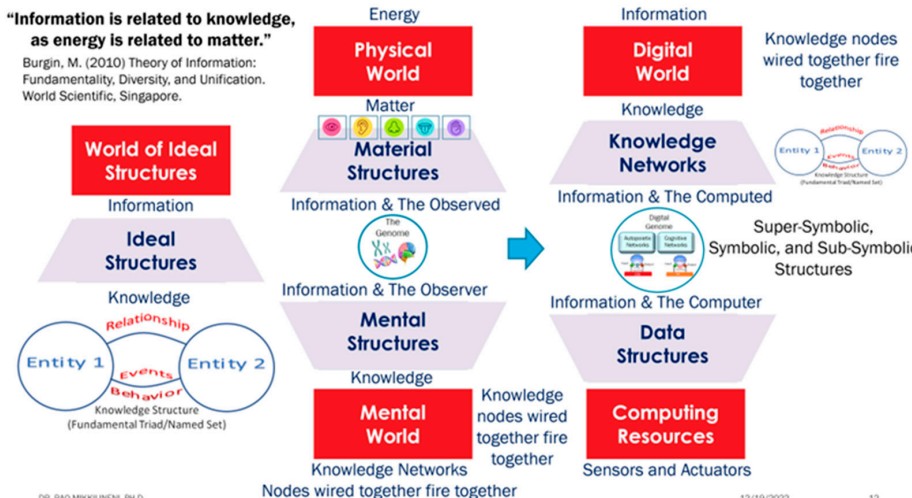

**Figure 2.** The role of information and knowledge in the material, mental, and digital worlds [1].

## 5. Digital Genome and Medical Knowledge-Based Digital Assistant Implementation

As Kelly et al. [25] point out, "the digital genome specifies the knowledge to execute various tasks that implement functional requirements, non-functional requirements, and the best practices to assure that the process objectives are achieved. Functional requirements deal with domain-specific process execution, components of knowledge acquisition, model creation, the evolution of knowledge structures, analysis of events and actions, etc. Non-functional requirements deal with deploying, operating, and managing the compute resources for the various application components, where to set up IaaS and PaaS (containers, databases, etc.), deploy auto-failover, auto-scaling, and live-migration policies. Best practices deal with decision-making by looking at the evolution of the system, predicting risk, and suggesting remedies for both maintaining functional and non-functional requirement fulfillment".

Figure 3 shows the process involved in creating the digital genome using various sources of knowledge. The autopoietic network manager deploys and regulates the hardware and software components to fulfill non-functional requirements and the cognitive manager self-regulates the workflow and process execution to fulfill functional requirements. The policies and best-practice knowledge are used to assure stability, safety, security, and survival while fulfilling the system's goal.

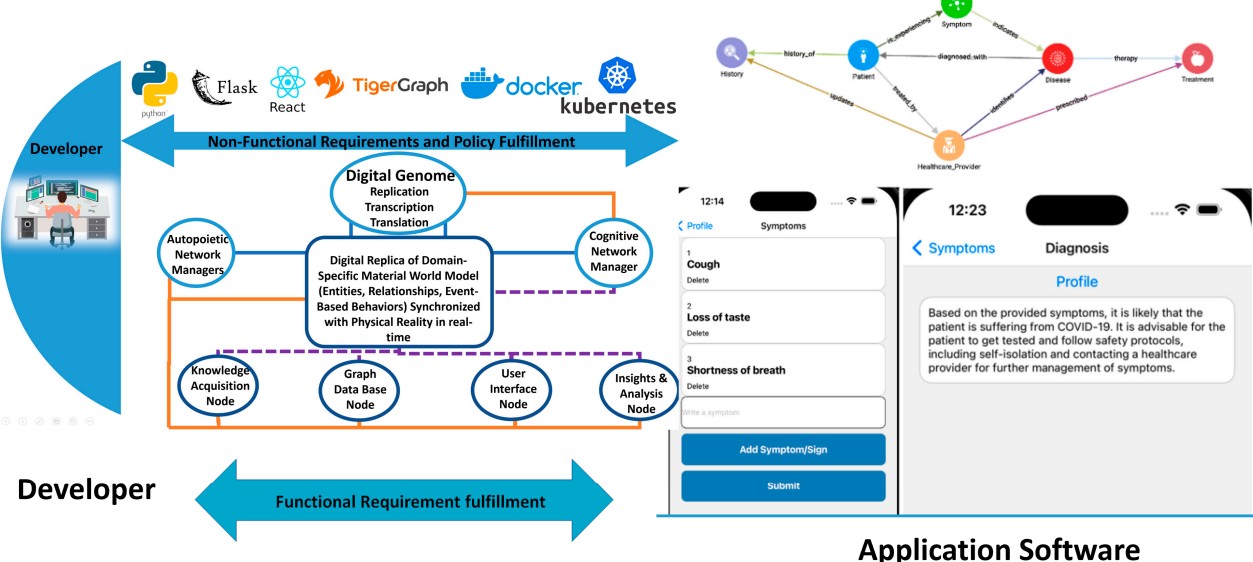

**Figure 3.** Architecture of medical knowledge-based digital assistant.

We refer the reader to the paper [25] which discusses the details and points to a demo of the system described.

## 6. Discussion

Clearly, Mark Burgin's general theory of information is a comprehensive and unified approach to the study of information. It provides a framework for understanding the nature of information, its various forms and manifestations, and the ways in which it can be measured and evaluated. The GTI is important because it offers a unified context for existing directions in information studies, making it possible to elaborate on a comprehensive definition of information, explain relations between information, data, and knowledge, and demonstrate how different mathematical models of information and information processes are related. The GTI also provides a new practically oriented perspective on information processes, including education, entertainment, and networking, and addresses problems of information dynamics and pragmatics. The GTI is based on a system of principles that explain what information is (by means of ontological principles) and how to measure information (by means of axiological principles). The theory also incorporates a parametric approach that provides a tool for building the GTI as a synthetic approach, which organizes and encompasses all main directions in information theory.

Roman Krzanowski [26] provides a comprehensive review of various theories of information and concludes the two theories, the general description of information (GDI) by Floridi [27] and the GTI by Burgin. He argues that the GTI appears to be the better of these two options because it is more fundamental and comprehensive with deep metaphysical roots. He concludes, "The GTI is certainly so fundamental that it creates a metaphysical basis for any type of information. If you want to regard information as a kind of Platonic essence, the GTI is the way to go. Nevertheless, the GDI complete as it is, is in large measure limited to semantic information. As comprehensive as such theories are, none of these theories have received the level of recognition attributed to Shannon's theory of communication (TOC) and its measure of information entropy. Shannon's TOC is operational and computable and has immense practical import, and it has inspired several extensions. It correlates with the Kolmogorov–Chaitin complexity theory and points to possible connections between information and fundamental physical laws, so it is a hard act to follow. In contrast, the GTI is a deep, philosophical, conceptual, foundational theory that tells us about foundational concepts and organizes dispersed conjectures into a whole, which is no easy task. Nevertheless, being highly theoretical, the GTI may not play as large a role as the TOC, except perhaps in the philosophy of information and in guiding our general comprehension of information. In reality, though, Shannon's TOC and Burgin's GTI are not competing theories. Using the metaphor of a car engine, we could say that Shannon's TOC tells us how the car engine is working, while Burgin's GTI tells us why it is working".

Hopefully, this paper demonstrates the potential practical implications of GTI that will profoundly influence future information systems.

## 7. Conclusions

Building information systems that mimic human intelligence is the holy grail of human thought. As Dodig-Crnkovic [28] points out, "When Turing discussed the possibility of constructing artificially intelligent agents based on computations performed by electronic machines, he was met with skepticism by many. Even to this day, human intelligence and especially consciousness are often considered impossible to implement in machines due to their supposed substantial differences in nature".

Current state-of-the-art machine intelligence uses the metabolism of computing hardware (using the power, CPU, and memory to execute computations) and the replication ability of software to mimic the process execution capability of a single cell and the capability of a collection of neurons (using an algorithm) to process information and convert it into knowledge. Figure 4 shows the current state-of-the-art in developing process automa-

tion and intelligent decision support systems using general-purpose computers. Data are converted into data structures and programs execute symbolic or sub-symbolic computing algorithms to process information and convert it into knowledge. The sub-symbolic computing structures provide the knowledge representation as parameters in a deep-learning algorithm and access to this knowledge is made available as information with generative pretrained transformer (GPT) programs. This information is used to create other knowledge representations (as the digital genome implementation demonstrates) and use other symbolic computing structures to provide process automation and intelligent support.

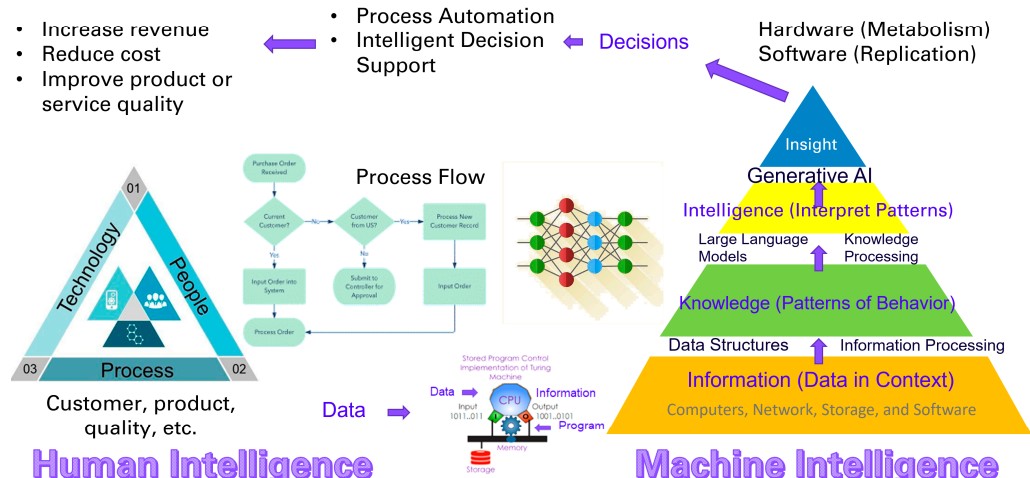

**Figure 4.** Machine intelligence used in business process automation and decision support (current state-of-the-art).

However, these symbolic, sub-symbolic, and hybrid computing structures still suffer from:

1.  The Church–Turing thesis limitations that deal with resource constraints [29] (p. 145).
2.  In her paper "Significance of Models of Computation, from Turing Model to Natural Computation", Gordana Dodig-Crnkovic states and justifies [30] that if a machine is composed of asynchronous concurrently running subsystems and their relative frequencies vary randomly, then such a machine cannot be adequately modeled by a Turing machine. Similarly, in her paper "Info-computationalism and Morphological Computing of Informational Structure", [30] she writes that while synchronous parallel processing can be made sequential and thus modeled by a Turing machine, asynchronous processes in networks cannot be appropriately modeled by a Turing machine [31].
3.  The lack of integration of the computer and the computed, with self-awareness of the system's purpose and the ability to self-regulate to accomplish that purpose [23,24].

In addition, an important implication of Gödel's incompleteness theorem [31] is that it is not possible to have a finite description with the description itself as the proper part. In other words, it is not possible to read yourself or process yourself as a process. However, as Turing [32,33] put it beautifully, "the well-known theorem of Gödel (1931) shows that every system of logic is in a certain sense incomplete, but at the same time, it indicates means whereby from a system L of logic a more complete system L_ may be obtained. By repeating the process we get a sequence L, L1 = L_, L2 = L_1 ... each more complete than the preceding. A logic L$\omega$ may then be constructed in which the provable theorems are the totality of theorems provable with the help of the logics L, L1, L2, ... Proceeding in this way we can associate a system of logic with any constructive ordinal. It may be asked whether such a sequence of logics of this kind is complete in the sense that to any problem A there corresponds an ordinal $\alpha$ such that A is solvable by means of the logic L$\alpha$".

These limitations are overcome with the super-symbolic structure with the specification of the digital genome [4,14,15]. The resulting autopoietic and cognitive behaviors provide the next big improvement in how we implement next-generation information systems which extend human intelligence by reducing the knowledge gap between actors using the system and help eliminate self-referential circularity not moored to external reality by synchronizing the mental models and external reality.

Figure 5 shows the new approach using GTI to define the digital genome and implement the knowledge networks using the structural machine where the knowledge structures wired together fire together to exhibit autopoietic and cognitive behaviors. The important differences between the current state and the GTI-based approach are:

1.  The genome specifies the system's functional requirements, non-functional requirements, and the best-practices-based policies.
2.  The structural machine implementation with autopoietic and cognitive knowledge networks provides operational resiliency and self-regulation to execute systems goals.
3.  Common knowledge representation integrating symbolic and sub-symbolic information processing structures provides a transparent and integrated view of the state and its dynamics in the form of history.
4.  Intelligent decision-making and process automation are accomplished using not only the current state but also the history and best practices where the future is predicted assessing the risk and mitigating it with higher-level reasoning that includes what-if simulations.

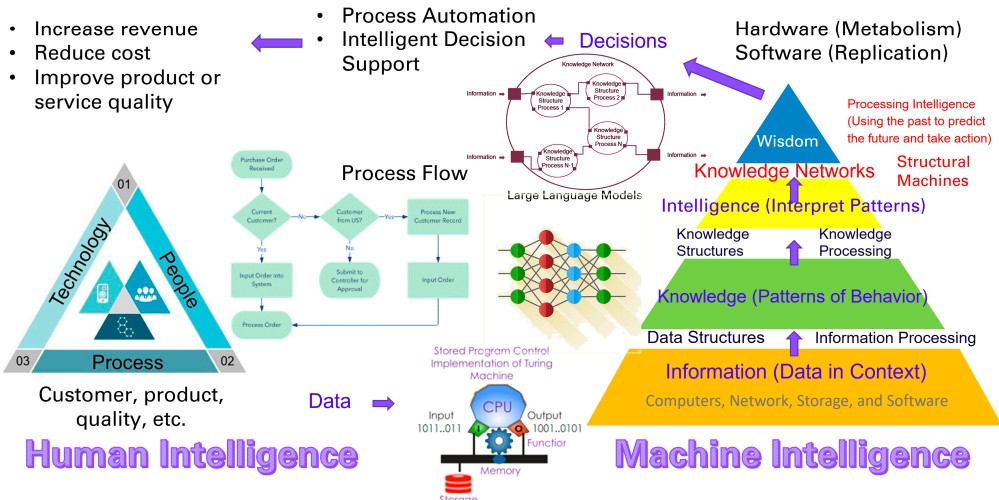

**Figure 5.** Schema-based approach using the structural machines (GTI-based).

Figure 6 summarizes the full cycle of state evolution of the system and prediction methodology.

Given the importance of the named sets in the general theory of information, we conclude this paper with a quote from Burgin [18]:

"It is also important to establish a consistent terminology in the area of AI. A terminology is a system of terms (names) used in a definite area, e.g., in science, art, business, or industry". The importance of the usage of correct names was stressed by the famous Chinese philosopher Confucius, who wrote:

"If names be not correct, language is not in accordance with the truth of things. If language be not in accordance with the truth of things, affairs cannot be carried on to success. When affairs cannot be carried on to success, proprieties and music do not flourish. When proprieties and music do not flourish, punishments will not be properly awarded. When punishments are not properly awarded, the people do not know how to move hand or foot".

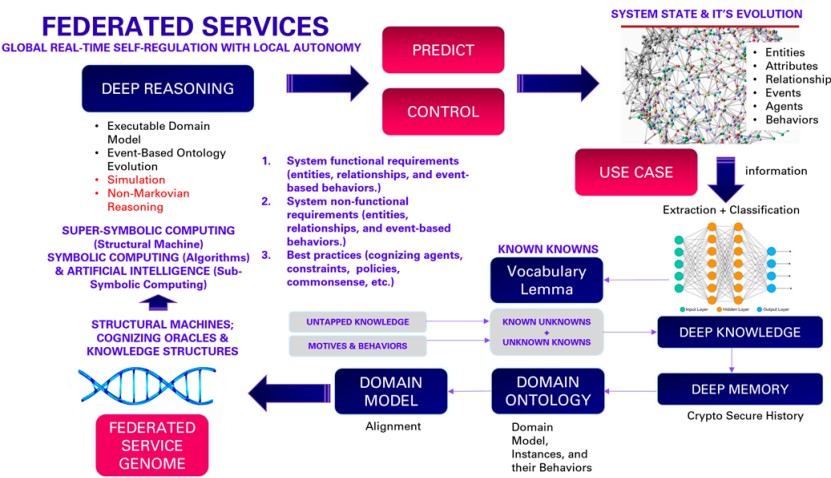

**Figure 6.** System state and evolution using the structural machines and knowledge networks.

**Funding:** This research received no external funding.

**Institutional Review Board Statement:** Not applicable.

**Informed Consent Statement:** Not applicable.

**Data Availability Statement:** Data are contained within the article.

**Conflicts of Interest:** The author declares no conflict of interest.

### Glossary

**Schema:** One of the most powerful and flexible forms of knowledge representations is the schema. The term "schema" has been used historically by philosophers like Immanuel Kant and later in neuroscience and psychology. A mental schema is both a mental representation (descriptive knowledge) of the world and operational knowledge that determines action in the world. Schemas are extensively utilized by people and computers for various purposes, including cognitive schemas, database schemas, and programming schema languages.

**Structural machines:** The structural machine is an information processing mechanism that works with schemas of knowledge structures and performs operations on them to evolve information changes in the system from one instant to another when any of the attributes of any of the objects change. The structural machines surpass the Turing machines, which work only with such primitive structures as strings of symbols, by their representations of knowledge and the operations that process information. Triadic structural machines with an assortment of general and mission-oriented processors and other triadic automata enable autopoietic behaviors.

**Knowledge networks:** Autopoietic machines are built using a knowledge network with knowledge nodes called knowledge structures that execute processes and carry information-sharing links between them. The knowledge nodes are wired together and fire together to manage the behavioral changes in the system. Each knowledge node contains hardware, software, and "infware" that manage the information processing and communication structures within the node. The infware of a system consists of diverse information carriers specifying how to discover, configure, monitor, and manage the hardware, software, and other software to maintain their state evolution based on externally infused knowledge such as business requirements dealing with system availability, performance, security, privacy, and regulatory compliance.

**Cognizing Oracles:** For Turing and the majority of computer scientists, an oracle is a device that supplies a Turing machine with the values of some function (on the natural numbers or words in some alphabet) that is not recursively, e.g., Turing machine, computable. This concept is generalized to define a software agent who has knowledge about a particular process and how to execute it securing the resources required.

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
