# Peer review of "Mark Burgin’s Legacy: The General Theory of Information, the Digital Genome, and the Future of Machine Intelligence"

_philosophies, doi:10.3390/philosophies8060107_

Round 1
Reviewer 1 Report
Comments and Suggestions for Authors
While rating the manuscript as Average my intent is to qualify this paper as a review of previous content by the authors.
Author Response
Thank you for qualifying the paper for publication.
Reviewer 2 Report
Comments and Suggestions for Authors
The paper gives an interesting detailed description of Mark Burgin's contribution to general theory of information (GTI). At the same time, since Mark Burgin was a mathematician, a significant part of his scientific results relate to pure mathematics and the problem of its foundation. For example, many of those results are described in joint publications with Marek Czachor and maybe it is reasonable to ask him to write an article about these results. I think that in this paper the author touched upon problems associated not only with GTI, but also with the interplay between GTI and foundation of pure mathematics. In particular, on page 10 the author discusses the importance of Gödel’s incompleteness theorems for GTI. However, it follows from those theorems that standard mathematics based on infinitesimals has its own foundational problems which cannot be resolved. For this reason, several authors consider approaches where physics is based on mathematics not containing infinitesimals. An important step in this direction was Mark Burgin's book
Burgin, M., & Czachor, M. (2020) "Non-Diophantine arithmetics in mathematics, physics and psychology". New York/London/Singapore: World Scientific
and paper
Burgin, M., Lev, F. "An Approach to Building Quantum Field Theory Based on Non-Diophantine Arithmetics." Foundations of Science (2023). https://doi.org/10.1007/s10699-022-09881-x .
Here it is shown that in problems where standard quantum theory with infinitesimals contains divergences, in the approach based on Burgin's Non-Diophantine Arithmetics, the expressions for Feynman diagrams become finite.
My main recommendation on the paper is to accept it in the present form, but, if the author thinks that my remarks may be useful, he/she can add his/her description of these problems.
Author Response
I would agree with the reviewer's main recommendation, which is to accept it in its present form. It is mainly because I am not a mathematician and am not equipped to review the subjects mentioned in the reviewer's comments. Perhaps the editors could invite Marek Czachor and F. Lev to contribute articles on the topics mentioned.
My focus has been the application of the General Theory of Information to build a new class of digital automata with autopoietic and cognitive behaviors which is a major advancement in how we build information systems that are resilient and provide assistance in making intelligent decisions in the context of business processes and their evolution. As mentioned in the paper, Mark Burgin's contribution will change the course of future AI.
Again, my thanks to the reviewer for suggesting to accept the paper, in its present form.